# Diagnostic Value of Circulating miR-202 in Early-Stage Breast Cancer in South Korea

**DOI:** 10.3390/medicina56070340

**Published:** 2020-07-09

**Authors:** Jungho Kim, Sunyoung Park, Dasom Hwang, Seung Il Kim, Hyeyoung Lee

**Affiliations:** 1Department of Biomedical Laboratory Science, College of Health Sciences, Catholic University of Pusan, Busan 46252, Korea; jutosa70@cup.ac.kr; 2Department of Biomedical Laboratory Science, College of Health Sciences, Yonsei University, Wonju 26493, Gangwon, Korea; angelsy88@gmail.com (S.P.); hdasom208@naver.com (D.H.); 3School of Mechanical Engineering, Yonsei University, Seoul 03772, Korea; 4Department of Surgery, College of Medicine, Yonsei University, Seoul 03772, Korea

**Keywords:** breast cancer, circulating biomarker, *miR-202*, RT-qPCR, diagnosis

## Abstract

*Background and objectives*: Breast cancer is the most common cancer among women worldwide. Early stage diagnosis is important for predicting increases in treatment success rates and decreases in patient mortality. Recently, circulating biomarkers such as circulating tumor cells, circulating tumor DNA, exosomes, and circulating microRNAs have been examined as blood-based markers for the diagnosis of breast cancer. Although *miR-202* has been studied for its function or expression in breast cancer, its potential diagnostic value in a clinical setting remains elusive and *miR-202* has not been investigated in South Korea. In this study, we aimed to evaluate the diagnostic utility of *miR-202* in plasma samples of breast cancer patients in South Korea. *Materials and Methods:* We investigated *miR-202* expression in the plasma of 30 breast cancer patients during diagnosis along with 30 healthy controls in South Korea by quantitative reverse transcription PCR. *Results:* The results showed that circulating *miR-202* levels were significantly elevated in the breast cancer patients compared with those in healthy controls (*p* < 0.001). The sensitivity and specificity of circulating *miR-202* were 90.0% and 93.0%, respectively. Additionally, circulating *miR-202* showed high positivity at early stage. The positive rate of *miR-202* was as follows: 100% (10/10) for stage I, 90% (9/10) for stage II, and 80% (8/10) for stage III. *miR-202* was also a predictor of a 9.6-fold high risk for breast cancer (*p* < 0.001). *Conclusions:* Additional alternative molecular biomarkers for diagnosis and management of pre-cancer patients are needed. Circulating *miR-202* might be potential diagnostic tool for detecting early stage breast cancer.

## 1. Introduction

Breast cancer is one of the common causes of cancer-mediated deaths among women worldwide. According to the World Health Organization (WHO), 1.5 million newly diagnosed cases and 570,000 deaths related to breast cancer were recorded in 2012. Breast cancer represents approximately 25% of total cancer cases and 15% of total deaths among women [1]. In the Republic of Korea, a total of 19,142 incidences of breast cancer with 2338 mortalities were reported in 2015 [2].

Despite continuous advancements in diagnosis methods, early diagnosis and improvement in survival are still difficult. The early detection and application of improved treatment are extremely important to improve the survival and quality of life for breast cancer patients. Along with simplicity, rapidness, and non-invasiveness, blood-based tumor markers are playing an increasingly important role in the diagnosis and treatment of breast cancer.

MicroRNAs (also known as miRNA or miR) are non-cording RNAs (19–25 nucleotides long) involved in the regulation of various cellular processes [3,4]. miRNAs bind to the 3′-untranslated regions (UTRs) of the target mRNAs, thereby either inhibiting their translation or inducing their sequence-specific degradation, leading to the silencing of respective gene expression. miRNAs regulating the expression of target mRNAs that promote tumor growth, invasion, angiogenesis, metastasis, and immune evasion have emerged as one of the major components in cancer biology [5,6,7,8,9]. miRNAs circulating in the blood have been identified, and their profiles reflect the variety of cancer types, suggesting that their circulating population is partly derived from cancer [10,11].

Recent studies have demonstrated that *miR-202* is associated with several types of cancers such as ovarian cancer [12], lung cancer [13], and colorectal cancer [14]. *miR-202* has been reported to have either a tumor-suppressive or oncogenic function (Table 1) [15,16,17,18,19,20,21,22,23,24,25,26,27,28].

This discrepancy is presumed to be the result of differences in the sample processing methods, type of samples, detection methods, and characteristics of the recruited study groups. Some studies showed that *miR-202* was significantly upregulated in breast cancer patients compared with that in healthy controls [24]. Additionally, it was highly expressed in the drug-resistant breast cancer tissues [25]. However, some studies reported that it was downregulated in breast cancer cells and inhibited the tumorigenesis of breast cancer [27,28]. Although *miR-202* has been studied for its function or expression in breast cancer, little is known about its potential diagnostic value in a clinical setting. *miR-202* has not been investigated in South Korea. Here, we evaluated the diagnostic utility of *miR-202* in plasma samples of breast cancer patients in South Korea. Furthermore, *miR-202*-based risk prediction for diagnosing breast cancer was investigated.

## 2. Materials and Methods

### 2.1. Clinical Samples

Plasma samples of 30 breast cancer patients (stages I–III) from the Department of Surgery, Yonsei Severance Hospital, Seoul, Republic of Korea, between 2011 to 2015 were used for *miR-202* expression analysis (Table 2). Data such as the age and Tumor-Node-Metastasis (TNM) stage of total 30 breast cancer patients were obtained from patients’ electrical medical records. For healthy control, plasma samples were obtained from 30 healthy donors. This study was approved by the institutional ethics committee at Yonsei Severance Hospital (approval no.: 4-2011-0011, date: 7 March 2011).

### 2.2. miRNA Extraction

For the extraction of miRNAs from plasma, the NuceloSpin^®^ miRNA Plasma kit (Macherely-Nagel, Düren, Germany) was used according to the manufacturer’s instructions. All the preparation and handling procedures were conducted under RNase-free conditions. Extracted miRNA was stored at −80 °C until used.

### 2.3. miRNA Expression Analysis

Complementary DNA (cDNA) was synthesized using the TaqMan microRNA Reverse Transcriptase kit (Applied Biosystems by Life Technologies, Foster City, CA, USA) according to the manufacturer’s recommendations. Briefly, 5 µL of miRNA was used for cDNA synthesis. Reverse transcriptase (RT) mixture contained 0.15 µL of 100 mM dNTP mix (100 mM of each dATP, dGTP, dCTP, and dTTP), 1 µL of 50 U/µL reverse transcriptase, 1.5 µL of 10× reverse transcriptase buffer, 0.19 µL of 20 U/µL RNase inhibitor, and 3 µL of miRNA-specific primer. The volume of RT mixture was adjusted to 15 µL with nuclease-free water. The following primers of TaqMan small RNA assays (Applied Biosystems by Life Technologies) were used: *hsa-miR-16* and *hsa-miR-202*. The temperature profile for cDNA synthesis reaction was: 16 °C for 30 min, 42 °C for 30 min, and 85 °C for 5 min.

The miRNA expression was quantified by determining the cycle threshold (C_T_), which is the number of PCR cycles required for the fluorescence to exceed a value significantly higher than the background fluorescence, using the TaqMan small RNA assays (Applied Biosystems by Life Technologies) with miRNA-specific primers according to the manufacturer’s instructions [29]. Briefly, 1.4 µL of cDNA was added to 10 µL of probe qPCR mix with 1 µL of miRNA-specific primer and 7.6 µL of nuclease-free water in the final volume of 20 µL. RT-qPCR reactions were performed on the CFX96 Real-Time PCR System Detector (Bio-Rad, Hercules, CA, USA). Samples were run in duplicate for each experiment. Data were analyzed using the comparative ΔΔC_T_ method (2^−ΔΔC^_T_) with miR-16 as an endogenous control for plasma [30,31,32]. To monitor contamination of the reagents, a negative control was included for each primer pair. PCR cycling conditions were as follows: 95 °C for 10 min, 40 cycles of 95 °C for 15 s, and 60 °C for 60 s.

### 2.4. Statistical Analyses

All the statistical analyses were performed using GraphPad Prism version 6.0 (La Jolla, CA, USA) and SPSS Statistics version 21.0 (IBM, Armonk, NY, USA). Student’s *t*-test was used for comparing *miR-202* expression between normal and cancer plasma samples, as well as to investigate *miR-202* expression in the patients according to TNM stage. To assess the diagnostic utility of *miR-202*, the receiver operating characteristic (ROC) curve analysis was performed and the area under the ROC curve (AUC) was calculated. The risk of breast cancer for miR-202 was analyzed by the chi-square test. Each stage of breast cancer was compared with healthy control. The effects were reported as odds ratios (ORs) and 95% confidence intervals (CIs). In all the analyses, *p* < 0.05 was considered statistically significant.

## 3. Results

### 3.1. Patients’ Characteristics

Blood samples were collected from 30 breast cancer patients who did not receive any anti-tumor treatment. The characteristics of the subjects involved in this study are shown in Table 2. The age of these 30 patients ranged from 31 to 76 years. There were 10 patients with stage I (33.3%), 10 with stage II (33.3%), and 10 with stage III (33.3%) breast cancer.

### 3.2. Diagnostic Value of Circulating miR-202

To investigate the diagnostic utility of *miR-202* in plasma samples, its expression levels were investigated by RT-qPCR, which were significantly higher in the breast cancer patients than in the healthy controls (*p* < 0.001; Figure 1a). The diagnostic performance of *miR-202* was determined by the analysis of ROC curve. The area under the ROC curve (AUC) of *miR-202* was 0.9500 (95% CI, 0.8842–1.016, *p* < 0.0001; Figure 1b). At the cut-off value of 2.1, the sensitivity, specificity, positive predictive value, and negative predictive value of *miR-202* were 90.0% (95% CI, 73.5–97.9), 93.3% (95% CI, 77.9–99.2), 90.3% (95% CI, 76.1–96.5), and 93.1% (95% CI, 77.9–98.1), respectively (Figure 1c).

### 3.3. Circulating miR-202 Expression According to TNM Stages

Subsequently, the expression levels of circulating *miR-202* were analyzed according to TNM stages, revealing a statistically significant difference between the expression level of each stage and healthy control (*p* < 0.001 for stage I, *p* < 0.001 for stage II, and *p* = 0.001 for stage III; Figure 2a). Importantly, the positivity rate was 95% (19/20) for early stage (stages I and II; Figure 2a).

To investigate the prediction power for diagnosis of breast cancer, the risk of breast cancer for *miR-202* was determined. We found that *miR-202* conferred a 4.3-fold (95% CI, 1.6–11.7, *p* < 0.001) at stage I, 3.9-fold (95% CI, 1.4–10.3, *p* < 0.001) at stage II, 3.4-fold (95% CI, 1.3–9.0, *p* < 0.001) at stage III, and 9.6 fold (95% CI, 3.3–28.3, *p* < 0.001) in total breast cancer patients (Figure 2b).

## 4. Discussion

The detection of circulating biomarkers can provide the understanding of their clinical implications and their potential use in a liquid biopsy for the diagnosis and treatment of cancer [33,34]. Among circulating biomarkers, circulating miRNAs are abundant in the circulatory system of blood and are resistant to RNase-mediated degradation. Circulating miRNAs are also stable in harsh conditions, including bipolar pH, extended storage, freeze–thaw cycles, and in formalin-fixed, paraffin-embedded tissue sections [35,36]. In this study, to evaluate circulating *miR-202* as a diagnostic tool for breast cancer, the performance characteristics of its sensitivity, specificity, PPV, and NPV were determined using plasma samples of the breast cancer patients.

In a previous study, the diagnostic potential of *miR-202* was determined in whole blood samples of 24 breast cancer patients and 24 healthy control using RT-qPCR [37]. *miR-202* was found to be highly expressed in breast cancer and the AUC for *miR-202* was 0.68. In addition, Joosse et al. [24] reported that the *miR-202* levels were elevated in a cohort of breast cancer patients (*n* = 102) compared with those in healthy controls (*n* = 37), which is identical to the results of this study. Our result showed that the sensitivity and specificity of circulating *miR-202* were 90.0% and 93.0%, respectively. The AUC value was 0.95 (95% CI = 0.88–1.02, *p* < 0.0001).

Numerous studies focus on improving or developing a treatment for improved prognosis; however, one of the most promising approaches is to detect the cancer at the early stage. The mammography and ultrasound are currently the standard diagnostic tools, proven to be useful for the detection of early stage breast cancer. However, the need for a new minimally invasive diagnostic approach is necessary to complement the mammography and improve the detection rates and breast cancer screening regulations. The specificity of mammography is more than 95%, but the sensitivity is between 67% and 95%, and is highly dependent on diverse factors such as age, breast density, and the professional experience of the investigator [38,39]. We found that *miR-202* was significantly upregulated in the plasma samples of early stage breast cancer (*p* < 0.001). Positivity rate was 100% (10/10) for stages I, 90% (9/10) for stage II, and 80% (8/10) for stage III. In addition, risk of breast cancer for *miR-202* was 9.6-fold in the plasma of breast cancer patients compared with that in the plasma of healthy controls.

Several studies have investigated the circulating miRNAs for breast cancer. Chen et al. showed that the expression levels of *miR-127-3p*, *miR-148b*, *miR-409-3p*, *miR-376a*, *miR-376c*, *miR-652*, and *miR-801* in the plasma of breast cancer patients were higher than the plasma of healthy controls. The diagnostic potential of these seven circulating miRs combined in 120 breast cancer patients showed an AUC value of 0.81 (95% CI 0.72–0.91) [40]. Additionally, Schrauder et al. reported that serum concentrations of *miR-106a-5p* and *miR454-3p* were higher in cancer patients than in healthy controls, whereas those of *miR-195-5p*, *miR-495*, and *miR-34a-5p* were downregulated in breast cancer patients. *miR-195-5p* and *miR-495* can help differentiate between breast cancer and healthy controls with AUC values (sensitivity, specificity) of 0.901 (77.8%, 100%) and 0.901 (100%, 66.7%), respectively [37]. The sensitivity and specificity of *miR-202* were comparable to those of other miRs for the breast cancer diagnostic method.

## 5. Conclusions

Our results show that *miR-202*-specific RT-qPCR may emerge as a useful tool in the diagnosis of breast cancer, and especially at its early stage. The limitation of this study was the small sample size from a single institution. Therefore, it is necessary to conduct additional studies using a large number of samples from the patients with various stages of breast cancer from multiple centers. Metastasis is one of the major causes of cancer-related deaths where early prediction can increase the survival rate. Investigation of the relationship between prognosis and the level of circulating *miR-202* is also necessary.

## Figures and Tables

**Figure 1 medicina-56-00340-f001:**
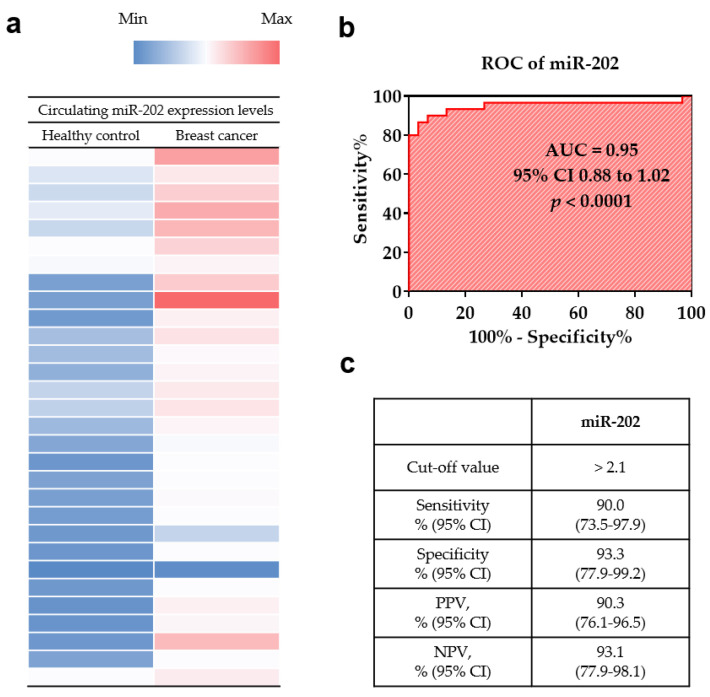
Diagnostic utility of circulating *miR-202* in plasma for breast cancer. (**a**) Heatmap of circulating *miR-202* in breast cancer patients and healthy controls. The color scheme is based on the gene expression level with upregulation in the red color and down regulation in the blue color. (**b**) The receiver operator characteristic curve (ROC) analysis of circulating *miR-202*. (**c**) Sensitivity, specificity, positive predictive value (PPV), and negative predictive value (NPV) of circulating *miR-202*.

**Figure 2 medicina-56-00340-f002:**
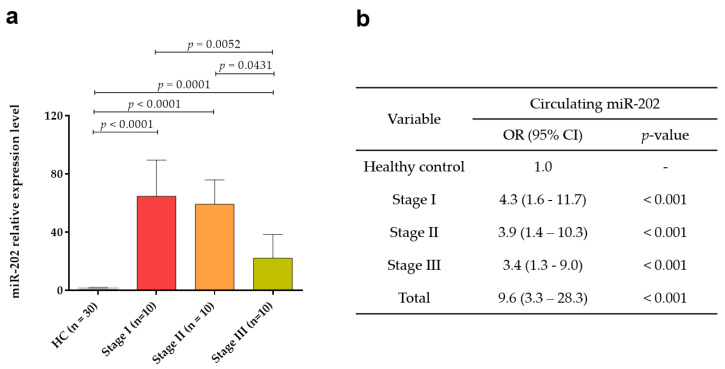
Expression levels of circulating *miR-202* according to Tumor-Node-Metastasis (TNM) stage. (**a**) The relative expression of circulating *miR-202* in plasma of breast cancer patients and healthy controls at different breast cancer stages (stages I, II, and III). (**b**) Odds ratios (ORs) of circulating *miR-202* according to TNM stage.

**Table 1 medicina-56-00340-t001:** Clinical characteristics of *miR-202* in tissue and blood samples.

	Cancer Type	Sample	Expression Level	Functions	Target	Reference
Tissue samples						
Chen et al. 2019	endometrial cancer	tissue	down	cell migration, invasion	FGF2	[15]
Wu et al. 2019	myocardial ischemia-reperfusion injury	mouse	up	cell apoptosis	TRPM6	[16]
Han et al. 2019	cervical cancer	tissue	up	cell migration, invasion, EMT	MALAT1	[17]
Ke et al. 2018	colorectal cancer	tissue	down	growth, metastasis	SMARCC1	[18]
Yang et al. 2017	glioma	tissue	down	growth, metastasis	MTDH	[19]
Jiang et al. 2016	lung cancer	tissue	down	cell cycle arrest, apoptosis	cyclin D1	[20]
Meng et al. 2016	esophageal squamous cell carcinoma	tissue	down	cell proliferation, migration	LAMA1	[21]
Wang et al. 2014	colorectal cancer	tissue	down	cell migration, proliferation	ARL5A	[22]
Blood samples						
Ma et al. 2016	esophageal squamous cell cancer	blood	down	cell migration, invasion	-	[23]
Joosse et al. 2014	Breast cancer	blood	up	metastasis, poor survival outcome	-	[24]

**Table 2 medicina-56-00340-t002:** Clinical characteristics of breast cancer patients.

Variable	No. of Cases	Percentage	Mean Age (Range)
TNM stage			
I	10	33.3	48.7 (34–60)
II	10	33.3	55.2 (31–76)
III	10	33.3	54.6 (38–73)

TNM, Tumor-Node-Metastasis.

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
