# Peer review of "Diagnostic Value of Circulating miR-202 in Early-Stage Breast Cancer in South Korea"

_medicina, 2020, doi:10.3390/medicina56070340_

Round 1
Reviewer 1 Report
In this Manuscript Kim et al., studied Mir202 is a diagnostic marker to screen early stage breast cancer in the clinical setup. Author investigated Mir-202 expression in the plasma samples of 30 breast cancer patients and 30 healthy controls in South Korea by quantitative RT- PCR analysis. Results showed elevated levels of mir202 in the patient samples compare to healthy individuals. My comments are very positive with minor suggestion that increasing sample size. Subject contains interesting and quality of the latest literatures were presented in the manuscript is good.
Author Response
Response to Reviewer 1 Comments
In this Manuscript Kim et al., studied Mir202 is a diagnostic marker to screen early stage breast cancer in the clinical setup. Author investigated Mir-202 expression in the plasma samples of 30 breast cancer patients and 30 healthy controls in South Korea by quantitative RT- PCR analysis. Results showed elevated levels of mir202 in the patient samples compare to healthy individuals.
Point 1: My comments are very positive with minor suggestion that increasing sample size. Subject contains interesting and quality of the latest literatures were presented in the manuscript is good.
Author’s response: We would like to appreciate for your time and careful review for our manuscript. As your comments, we mentioned limitation of small sample size from a single institution in discussion section. (Page 6, line 189-191 in revised manuscript)
Reviewer 2 Report
The research paper entitled "Diagnostic value of circulating miR-202 in early-stage breast cancer in South Korea" of Jungho Kim et al. analyzes the miR-202 expression on 30 plasma sample from breast cancer south Korean patients, showing that circulating miR-202 levels are significantly elevated in the early stage of breast cancer compared with healthy donors samples. The article is interesting despite the small sample size because it suggests new investigations about a possible role of miR-202 as a circulating biomarker for early detection of breast cancer.
Below, my comments and suggestions for authors:
- 3: miRNA reference used in the introduction is not adequate. It is better to change and insert more important and significant papers for this crucial point.
- References 4/5: also, in this case, it is better to insert a recent review about microRNA and cancer which explains in detail the biogenesis, role, functions, and regulation of miRNAs in cancer.
- In general, symbols for human microRNA are italicized. (hsa-mir-202)
- Row 56: the sentence “miR-202 has been reported to have either tumor suppressive or oncogenic function” needs a reference.
- Row 59: the sentence “some studies reported 58 that it was downregulated in breast cancer cells and inhibited the tumorigenesis of breast cancer” needs a reference.
- Row 70: “Data such as age and TNM stage of a total of 30 breast cancer patients were retrospectively reviewed from patients’ electrical medical records.” So, what it means “retrospectively reviewed”? The sentence is confusing. To select the sample size for the analysis the stage information is necessary at the beginning of the study.
- Row 71: “…30 healthy donors who had never been diagnosed with breast cancer”. In any case, a healthy donor is selected because healthy… so this sentence is unnecessary and confusing.
- Table 1: It is better to indicate the age of patients between tumor stages, in this way this information may be interesting.
- Insert primers sequences used in the study.
- Row 134: The method used for calculating the relative risk needs more details.
- Discussion chapter: in the discussion chapter there is first, the explanation of data obtained in the study and second, the linked literature data discussion, but generally it is better the contrary: first, literature data and second the discussion about the result obtained in the present work, that in this way, it is very clear.
- Row 166: “Positivity rate was 95% (19/20) for stages I and II.” The percentage is referred to Stage I and II together, why? And what are the values for the single tumor stage?
- Table 2: “Clinical characteristics of miR-202 in tissue and blood samples”. It is better to move table 2 on the introduction chapter, as well as the relative discussion (Row 180-184), and not as the final of the discussion chapter.
Author Response
Response to Reviewer 2 Comments
The research paper entitled "Diagnostic value of circulating miR-202 in early-stage breast cancer in South Korea" of Jungho Kim et al. analyzes the miR-202 expression on 30 plasma sample from breast cancer south Korean patients, showing that circulating miR-202 levels are significantly elevated in the early stage of breast cancer compared with healthy donors samples. The article is interesting despite the small sample size because it suggests new investigations about a possible role of miR-202 as a circulating biomarker for early detection of breast cancer.
Author’s response: We deeply appreciate for your time and careful review. Based on your comments, we have revised the manuscript accordingly.
Below, my comments and suggestions for authors:
Point 1: 3: miRNA reference used in the introduction is not adequate. It is better to change and insert more important and significant papers for this crucial point.
Author’s response: We changed miRNA references “Bartel, D.P. microRNAs: genomics, biogenesis, mechanism, and function. Cell. 2004, 116, 281-297 and Bartel, D.P. MicroRNAs: target recognition, and regulatory functions. Cell. 2009, 136, 215-233.”
- Bartel, D.P. MicroRNAs: genomics, biogenesis, mechanism, and function. Cell. 2004, 116, 281-297.
- Bartel, D.P. MicroRNAs: target recognition, and regulatory functions. Cell. 2009, 136, 215-233.
Point 2: References 4/5: also, in this case, it is better to insert a recent review about microRNA and cancer which explains in detail the biogenesis, role, functions, and regulation of miRNAs in cancer.
Author’s response: We added recent review about the biogenesis, role, functions, and regulation of miRNAs in cancer (reference 5-9)
- Anastasiadou E., Faggioni A., Trivedi P., Slack F.J. The nefarious nexus of noncoding RNAs in cancer. 2018, 19, 2072.
- Rupaimoole. R.; Salck, F.J. MicroRNA therapeutics: towards a new era for the management of cancer and other diseases. Nat. Rev. Drug Discov. 2017, 16, 203-222.
- Lin S., Gregory R.I. MicroRNA biogenesis pathways in cancer. Nat. Rev. Cancer. 2015, 15, 321-333.
- Stahlhut, C.; Slack, F.J. MicroRNAs and the cancer phenotype: profiling, signatures and clinical implications. Genome Med. 2013, 5, 111.
- Kasinski, A.L.; Slack, F.J. Epigenetics and genetics. MicroRNAs en route to the clinic: progress in validating and targeting microRNAs for cancer therapy. Nat. Rev. Cancer 2011, 11, 849–864
Point 3: In general, symbols for human microRNA are italicized. (hsa-mir-202)
Author’s response: We corrected symbols for human microRNA in italics
Point 4: Row 56: the sentence “miR-202 has been reported to have either tumor suppressive or oncogenic function” needs a reference.
Author’s response: We added reference about the sentence “miR-202 has been reported to have either tumor suppressive or oncogenic function” (reference 15-28)
- Chen, P.; et al. MicroRNA-202 inhibits cell migration and invasion through targeting FGF2 and inactivating Wnt/beta-catenin signaling in endometrial carcinoma. Biosci. Rep. 2019, 39, BSR20190680.
- Wu, H.Y.; Wu, J.L.; Ni, Z.L. Overexpression of microRNA-202-3p protects against myocardial ischemia-reperfusion injury through activation of TGF-beta1/Smads signaling pathway by targeting TRPM6. Cell Cycle 2019, 18, 621–637.
- Han, X.; et al. Long non-coding RNA metastasis-associated lung adenocarcinoma transcript 1/microRNA-202-3p/periostin axis modulates invasion and epithelial-mesenchymal transition in human cervical cancer. J. Cell Physiol. 2019, 234, 14170–14180.
- Ke, S.B.; et al. MicroRNA-202-5p functions as a tumor suppressor in colorectal carcinoma by directly targeting SMARCC1. Gene 2018, 676, 329–335.
- Yang, J.; et al. MicroRNA-202 inhibits cell proliferation, migration and invasion of glioma by directly targeting metadherin. Oncol. Rep. 2017, 38, 1670–1678.
- Jiang, J.; et al. MicroRNA-202 induces cell cycle arrest and apoptosis in lung cancer cells through targeting cyclin D1. Eur. Rev. Med. Pharmacol. Sci. 2016, 20, 2278–2284.
- Meng, X.; et al. MicroRNA-202 inhibits tumor progression by targeting LAMA1 in esophageal squamous cell carcinoma. Biochem. Biophys. Res. Commun. 2016, 473, 821–827.
- Wang, Q.; et al. microRNA-202-3p inhibits cell proliferation by targeting ADP-ribosylation factor-like 5A in human colorectal carcinoma. Clin. Cancer Res. 2014, 20, 1146–1157.
- Ma, G.; et al. Low expression of microRNA-202 is associated with the metastasis of esophageal squamous cell carcinoma. Exp. Ther. Med. 2016, 11, 951–956.
- Joosse, S.A.; et al. Circulating cell-free cancer-testis MAGE-A RNA, BORIS RNA, let-7b and miR-202 in the blood of patients with breast cancer and benign breast diseases. Br. J. Cancer 2014, 111, 909–917.
- Liu, T.; Guo, J.; Zhang, X. MiR-202-5p/PTEN mediates doxorubicin-resistance of breast cancer cells via PI3K/Akt signaling pathway. Cancer Biol. Ther. 2019, 20, 989–998.
- Fang, R.; et al. Plasma microRNA pair panels as novel biomarkers for detection of early stage breast cancer. Front. Physiol. 2018, 9, 1879.
- Gao, S.; et al. miR-202 acts as a potential tumor suppressor in breast cancer. Oncol. Lett. 2018, 16, 1155–1162.
- Xu, F.; Li, H.; Hu, C. MiR-202 inhibits cell proliferation, invasion, and migration in breast cancer by targeting ROCK1 gene. J. Cell Biochem. 2019, 120, 16008–16018.
Point 5: Row 59: the sentence “some studies reported 58 that it was downregulated in breast cancer cells and inhibited the tumorigenesis of breast cancer” needs a reference.
Author’s response: We added reference about the sentence “some studies reported that it was downregulated in breast cancer cells and inhibited the tumorigenesis of breast cancer” (reference 27, 28).
- Gao, S.; et al. miR-202 acts as a potential tumor suppressor in breast cancer. Oncol. Lett. 2018, 16, 1155–1162.
- Xu, F.; Li, H.; Hu, C. MiR-202 inhibits cell proliferation, invasion, and migration in breast cancer by targeting ROCK1 gene. J. Cell Biochem. 2019, 120, 16008–16018.
Point 6: Row 70: “Data such as age and TNM stage of a total of 30 breast cancer patients were retrospectively reviewed from patients’ electrical medical records.” So, what it means “retrospectively reviewed”? The sentence is confusing. To select the sample size for the analysis the stage information is necessary at the beginning of the study.
Author’s response: Based on the reviewer’s comments, we removed the sentence “retrospectively reviewed” to avoid the confusion for future readers and revised the sentence as follow.
(Page 3, line 73-74 in revised manuscript)
Data such as age and TNM stage of total 30 breast cancer patients were retrospectively reviewed from patients’ electrical medical records.
à Data such as age and Tumor-Node-Metastasis (TNM) stage of total 30 breast cancer patients were obtained from patients’ electrical medical records.
Point 7: Row 71: “…30 healthy donors who had never been diagnosed with breast cancer”. In any case, a healthy donor is selected because healthy… so this sentence is unnecessary and confusing.
Author’s response: To avoid the confusion for future readers, we removed the sentence ‘who had never been diagnosed with breast cancer’
(Page 3, line 74-75 in revised manuscript)
Point 8: Table 1: It is better to indicate the age of patients between tumor stages, in this way this information may be interesting.
Author’s response: we revised Table 1 to indicate the age of patents with tumor stages.
(Page 3, line 77 in revised manuscript, Table 2)
Point 9: Insert primers sequences used in the study.
Author’s response: We used TaqMan microRNA Reverse Transcriptase kit (Applied Biosystems by Life Technologies, Foster City, CA, USA) and TaqMan small RNA assays (Applied Biosystems by Life Technologies) to analysis microRNA expression. So, we added ‘Applied biosystems by Life Technologies’ site as reference which exist manual of TaqMan small RNA assays (reference 29).
- Applied Biosystems by Life Technologies. (https://www.thermofisher.com/order/genome-database/details/mirna/002363?pluginName=&CID=&ICID=)
Point 10: Row 134: The method used for calculating the relative risk needs more details.
Author’s response: As suggested by the reviewer, the calculation method of the risk of breast cancer for miR-202 described in material and method in detail. Based on the chi-square test, odds ratio analysis between healthy controls and stages of breast cancer were performed by Prism software 6.0.
(Page 4, line 110-112; Page 5, line 139 and line 146; Page 6, line 173 in the revised manuscript)
Point 11: Discussion chapter: in the discussion chapter there is first, the explanation of data obtained in the study and second, the linked literature data discussion, but generally it is better the contrary: first, literature data and second the discussion about the result obtained in the present work, that in this way, it is very clear.
Author’s response: As suggested, we revised discussion chapter (first, literature data and second the discussion about the results from this study).
(Page 5, line 154-163 in revised manuscript)
Point 12: Row 166: “Positivity rate was 95% (19/20) for stages I and II.” The percentage is referred to Stage I and II together, why? And what are the values for the single tumor stage?
Author’s response: As suggested, we described the positivity rate for stage I, II, and III, respectively. Positivity rate was 100% (10/10) for stages I, 90% (9/10) for stage II, and 80% (8/10) for stage III.
(Page 6, line 172-173 in revised manuscript)
Point 13: Table 2: “Clinical characteristics of miR-202 in tissue and blood samples”. It is better to move table 2 on the introduction chapter, as well as the relative discussion (Row 180-184), and not as the final of the discussion chapter.
Author’s response: We moved table 2 and sentence related to table 2 on the introduction section.
(Page 2, line 55-60 in revised manuscript, Table 1)
Reviewer 3 Report
The research article by Jungho Kim and colleagues, describes the diagnostic value of miR-202 detection in the plasma of breast cancer (BC) patients in South Korea. This review is beautifully written and timely. Although the authors are not the first who claim the diagnostic value of this miRNA in BC, still it is interesting to reconfirm this with their data, regardless of the small cohort in 30 patients from S. Korea. In addition, miR-202 might be used as an early stage diagnostic marker in BC.
I have a few minor points:
1) The authors could explain the acronym: TNM
2) The quality of figures could be adjusted.
3) The authors might want to add more recent references about the role of non-coding RNA in cancer, such as PMID: 30018188 and PMID: 28209991
Author Response
Response to Reviewer 3 Comments
The research article by Jungho Kim and colleagues, describes the diagnostic value of miR-202 detection in the plasma of breast cancer (BC) patients in South Korea. This review is beautifully written and timely. Although the authors are not the first who claim the diagnostic value of this miRNA in BC, still it is interesting to reconfirm this with their data, regardless of the small cohort in 30 patients from S. Korea. In addition, miR-202 might be used as an early stage diagnostic marker in BC.
Author’s response: We would like to appreciate for your time and careful review for our manuscript.
I have a few minor points:
Point 1: The authors could explain the acronym: TNM
Author’s response: We explained the acronym for TNM in materials and methods section. (Page 3, line 73 in revised manuscript)
Point 2: The quality of figures could be adjusted.
Author’s response: Based on the reviewer’s comments, we adjusted to a high quality of figures
Point 3: The authors might want to add more recent references about the role of non-coding RNA in cancer, such as PMID: 30018188 and PMID: 28209991
Author’s response: As suggested, we added recent references about role of non-coding RNA in cancer (reference 3, 4, 5, 6, 8, 9)
- Bartel, D.P. MicroRNAs: genomics, biogenesis, mechanism, and function. Cell. 2004, 116, 281-297.
- Bartel, D.P. MicroRNAs: target recognition, and regulatory functions. Cell. 2009, 136, 215-233.
- Anastasiadou E., Faggioni A., Trivedi P., Slack F.J. The nefarious nexus of noncoding RNAs in cancer. 2018, 19, 2072.
- Rupaimoole. R.; Salck, F.J. MicroRNA therapeutics: towards a new era for the management of cancer and other diseases. Nat. Rev. Drug Discov. 2017, 16, 203-222.
- Lin S., Gregory R.I. MicroRNA biogenesis pathways in cancer. Nat. Rev. Cancer. 2015, 15, 321-333.
- Stahlhut, C.; Slack, F.J. MicroRNAs and the cancer phenotype: profiling, signatures and clinical implications. Genome Med. 2013, 5, 111.
- Kasinski, A.L.; Slack, F.J. Epigenetics and genetics. MicroRNAs en route to the clinic: progress in validating and targeting microRNAs for cancer therapy. Nat. Rev. Cancer 2011, 11, 849–864
Round 2
Reviewer 2 Report
In my opinion, the authors have properly revised the manuscript.